# Predicting the Bioclimatic Habitat Suitability of *Ginkgo biloba* L. in China with Field-Test Validations

**Ying Guo [1,2], Jing Guo [1,2], Xin Shen [1,2], Guibin Wang [1,2,*] and Tongli Wang [3,*]**

[1] Co-Innovation Centre for Sustainable Forestry in Southern China, Nanjing Forestry University, Nanjing 210037, China

[2] College of Forestry, Nanjing Forestry University, Nanjing 210037, China

[3] Department of Forest and Conservation Sciences, University of British Columbia, Vancouver, BC V6T 1Z4, Canada

* Correspondence: gbwang@njfu.com.cn (G.W.); tongli.wang@ubc.ca (T.W.)

**Abstract:** Ginkgo (*Ginkgo biloba* L.) is not only considered a 'living fossil', but also has important ecological, economic, and medicinal values. However, the impact of climate change on the performance and distribution of this plant is an increasing concern. In this study, we developed a bioclimatic model based on data about the occurrence of ginkgo from 277 locations, and validated model predictions using a wide-ranging field test (12 test sites, located at the areas from 22.49° N to 39.32° N, and 81.11° E to 123.53° E). We found that the degree-days below zero were the most important climate variable determining ginkgo distribution. Based on the model predictions, we classified the habitat suitability for ginkgo into four categories (high, medium, low, and unsuitable), accounting for 9.29%, 6.09%, 8.46%, and 76.16% of China's land area, respectively. The ANOVA results of the validation test showed significant differences in observed leaf-traits among the four habitat types ($p < 0.05$), and importantly the rankings of the leaf traits were consistent with our classification of the habitat suitability, suggesting the effectiveness of our classification in terms of biological and economic significance. In addition, we projected that suitable (high and medium) habitats for ginkgo would shrink and shift northward under both the RCP4.5 and RCP8.5 climate change scenarios for three future periods (the 2020s, 2050s, and 2080s). However, the area of low-suitable habitat would increase, resulting in a slight decrease in unsuitable habitats. Our findings contribute to a better understanding of climate change impact on this plant and provide a scientific basis for developing adaptive strategies for future climate.

**Keywords:** *Ginkgo biloba* L.; climate change; habitat suitability; bioclimatic model; MaxEnt

## 1. Introduction

Ginkgo (*Ginkgo biloba* L.) is one of the oldest and most intriguing gymnosperms, probably emanating from the Permian (approximately 200 million years ago), and it is the only living species in the Ginkgophyta division [1]. Ginkgo is a popular tree native to China with high ecological, ornamental, medical, and scientific value [2–5]. By the end of the last century, the area of ginkgo plantations had reached nearly 200,000 hectares in China, and the cultivation of ginkgo has become a major source of income in some rural areas [6]. However, to the best of our knowledge, for such an important economic tree species, the impact of climate change on habitat suitability of ginkgo remains unclear. A changing climate is already exerting pressure on the plantation industry; many tree species continue to decline in numbers and their habitat suitability are projected to significantly decline [7–9]. Therefore, how ginkgo will respond to the challenges of climate change has become one of the most important issues and an increasing concern to plantation managers.

The rate of global climate change in the 21st century is unprecedented [10,11]. Elevated temperatures, changes in rainfall frequency, and frequent droughts can have a negative impact on tree growth and may lead to species extinction and loss of biodiversity [12,13]. Adaptation, extinction, and migration are considered three basic ways for trees to cope with climate change [14,15]. For ginkgo trees, the ability to migrate over long distances is limited by their physiological characteristics. First, the long-lived tree will be unlikely to migrate fast enough to avoid the negative effects of climate change. Second, the characteristics of shade intolerance, late reproductive maturity (approximately 20 years), clonal propagation of secondary trunks, and large seeds may be unfavorable to the migration of ginkgo populations [16–19]. Presumably, ginkgo may be in danger of extinction in some habitats if it fails to adapt to rapid climate change in the future. Thus, when vigorously developing the industry of ginkgo plantation, it is urgent to fully evaluate and quantify the potential risks of losing habitats for ginkgo under climate change.

In recent years, bioclimatic modeling has become an effective tool for assessing the constraints of species habitat suitability under climate change. These models use the relationship between known occurrences and absences with climate variables to predict habitat suitability [20]. Bioclimatic modeling adopts a general view that the best indicator of the climate needs of a species is its current distribution [21,22]. Although a bioclimatic model describes the suitability of ecological space, it is usually projected into geospatial space, resulting in a geographic region of predicted presence for the species. For example, these methods have been used in identifying and mapping the geographical distribution of potentially suitable climatic habitat of *Thuja sutchuenensis* Franch. [23], *Tilia amurensis* Rupr., *Phellodendron amurense* Rupr., *Chosenia arbutifolia* (Pallas) A. K. Skv., *Pinus koraiensis* Siebold et Zuccarini, *Pinus Sylvestris* Linn., *Fraxinus mandshurica* Rupr., and *Juglans mandshurica* Maxim. [24]. Despite the high consistency among bioclimatic models, many studies have shown that maximum entropy (MaxEnt) modeling is probably the most widely used and generally provides good discriminatory ability within the modeled region [25,26]. The MaxEnt algorithm has the following advantages: (1) Modeling requires only a species' presence-background data and environmental information throughout the study area [27]; (2) the algorithm has a concise mathematical definition, which is guaranteed to converge to the optimal (maximum entropy) probability distribution [28]; and (3) MaxEnt can output the relative contribution of each variable and directly generate spatially explicit habitat suitability maps [29–31].

Remarkably, despite a rapidly increasing volume of literature on bioclimatic models, there is relatively scarce evidence to support the prediction results of these models. The MaxEnt model represents a species' niche by associating known distributional information with suites of climate variables [28]. The climatic inputs used are considered to have direct physiological roles in limiting the ability of plants to survive. Although fundamentally correlative, the climate niche model combines an understanding of species' ecology and physiology [32]. Therefore, the physiological variations of plants under different climate conditions can be used to indirectly support the reliability of model outputs. Physiologically, the capacity of trees to successfully cope with climate change critically depends on their phenotypic plasticity—a physiological process that adapts to warming by changing their phenotypic characteristics [33]. Many ecologists believe that among phenotypic characteristics of trees, leaf-traits are most closely related to plant growth strategies and the ability of plants to utilize resources, which could reflect the survival strategies of plants adapted to environmental changes [34,35]. Different climatic conditions impose different selective forces on trees and drove leaf-traits to a certain degree of divergence [36]. These divergences might successfully link spatial habitat suitability to temporal climatic changes [37]. In addition, leaf production is directly relevant to its economic values for ginkgo because its leaves possess considerable medicinal benefits that are widely used in the field of medicine [38]. In China, the area of plantations with ginkgo leaves as the major product has exceeded 50,000 hectares, covering five climatic zones: temperate, warm temperate, north subtropical, mid-subtropical, and south subtropical zones [6]. Therefore, we conducted field tests nationwide to enable ginkgo with a controlled genetic background to be planted under different climatic conditions.

The response of ginkgo to climate change was evaluated by observing the plasticity of several leaf traits. The variations of the leaf-trait plasticity in response to climate among different ecological zones were used to validate the reliability of the bioclimatic model.

The main objective of this study was to evaluate the response of *Ginkgo biloba* to current and future climates using a bioclimatic model validated by a field test. Specifically, through model building, validation and predictions, we were to: (1) Determine the key climatic factors driving the ginkgo distribution; (2) classify habitats of ginkgo in China into ecological types based on climate suitability; and (3) predict changes in the habitat suitability of ginkgo under future climate change scenarios in this century (2011–2100). We anticipate that the results of this study would provide a scientific basis for economic plantation planning and for developing adaptive strategies of ginkgo for a changing climate.

## 2. Materials and Methods

### 2.1. Occurrence Data

We collected 1044 occurrence sample locations from geographical occurrence information of ginkgo from the Ginkgo Gene Bank of Nanjing Forestry University (the germplasm resource survey was carried out from 1998–2000). To reduce sampling biases, we partitioned the distribution area into 4 km × 4 km pixels, and only one sample point was selected in each grid, resulting in 277 locations used, as shown in Figure 1. Although the 4 km x 4 km rule is arbitrary, the process of spatial filtering is to satisfy the above goals without excessively reducing the number of distribution points (>100) [39,40].

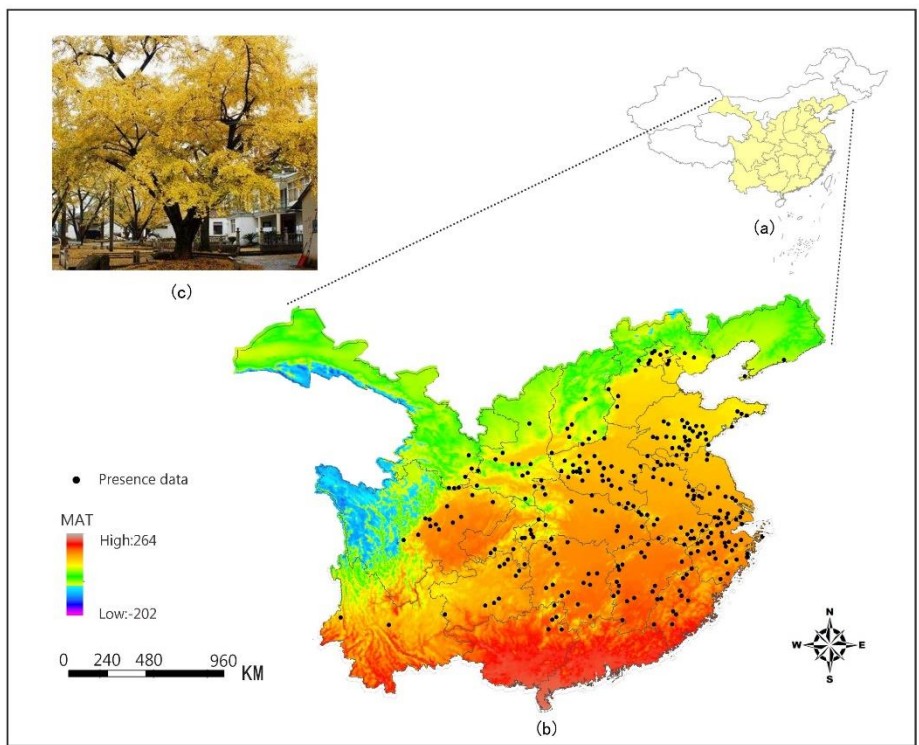

**Figure 1.** (**a**) Map of China; (**b**) distribution of the ginkgo presence points in major provinces and the different colors on the map reflect the changes of Mean Annual Temperature (MAT); and (**c**) *Ginkgo biloba* L.

### 2.2. Climate Data

To develop the model and generate raster files for spatial mapping, we used ClimateAP software (Version 2.03, available at [41]) to obtain climate variables [42]. For spatial mapping, we used an 800 m × 800 m digital elevation model (DEM) to prepare the input file to generate current (1960–1990)

and future (2011–2100) gridded climate data. In the ClimateAP software, climate data can be extracted from any location based on longitude, latitude, and altitude, and the output data covered the entirety of China.

The climate data for future periods, including the 2020s (2011–2040), 2050s (2041–2070), and 2080s (2071–2100), were obtained from general circulation models (GCMs) of the Coupled Model Intercomparison Project (CMIP5) included in the Intergovernmental Panel on Climate Change (IPCC) Fifth Assessment Report [13]. The representation concentration pathways (RCPs) 4.5 and 8.5 were selected in this study, where RCP4.5 is relatively optimistic and RCP8.5 is the scenario with the highest greenhouse gas emissions. The ensembles based on 15 GCMs for RCP4.5 and RCP8.5 included in ClimateAP were used for projections for the three periods.

Sixteen climate variables related to the physiological characteristics of ginkgo were selected (Table 1). To avoid strong correlations among the climate variables, we used Pearson correlation coefficients to measure pairwise correlations among the variables (R, Version 3.5.1), and one of the two paired variables correlated above 0.8 was eliminated.

**Table 1.** Climate variable code and its full name.

| Code | Full Name (Units) | Code | Full Name (Units) |
|:---:|:---:|:---:|:---:|
| MAT | Mean annual temperature (°C) | DD > 5 | Degree-days above 5 °C, growing degree-days (°C-days) |
| MWMT | Mean warmest month temperature (°C) | DD < 0 | Degree-days below 0 °C, chilling degree-days (°C-days) |
| MCMT | Mean coldest month temperature (°C) | NFFD | The number of frost-free days (day) |
| TD | Temperature difference between MWMT and MCMT, or continentality (°C) | PAS | Precipitation as snow between August in previous year and July in the current year (mm) |
| MAP | Mean annual precipitation (mm) | EMT | Extreme minimum temperature over 30 years (°C) |
| EXT | Extreme maximum temperature over 30 years (°C) | Eref | Hargreaves reference evaporation |
| AHM | Annual heat moisture index (MAT + 10)/(MAP/1000) | CMD | Hargreaves climatic moisture deficit |

*2.3. Model Development*

We used the maximum entropy approach to model the relationship between the species occurrence and climate variables. This approach performs well compared to many other bioclimatic models [43]. We used the MaxEnt 3.4.1 version and established the following settings in the software interface: check the Jackknife test; random test = 25%; regularization multiplier = 1; maximum background points = 10,000; and a repeat was run 10 times. We randomly divided the data into a training set (75%) for model development and a validation set (25%) to assess model performance. To consider the uncertainty introduced by splitting the training and validation sets, 10 models were built by 10 repeated runs for cross-validation. The remaining settings were kept as default. This setup has been considered to be reasonable and effective in a wide range of niche studies. MaxEnt assigns a non-negative probability to each pixel in the study area (probabilities sum to 1). To be easier to use and interpret, MaxEnt outputs the 'cumulative' probability for each pixel on the scale of 0%–100%. Therefore, the habitat suitability is classed by setting decision thresholds [28]. According to previous studies [27,44,45], we classified habitat suitability into four levels: (1) High-suitable habitat—probability values greater than 0.6; (2) medium-suitable habitat—values ranging from 0.4 to 0.6; (3) low-suitable habitat—values ranging from 0.1 to 0.4; and (4) unsuitable habitat—values less than 0.1.

The inbuilt model quality assessment routines of MaxEnt were used to assess the accuracy of the modeling prediction. First, through ten iterations, we obtained the standard deviation between the models to assess possible bias due to data splitting. We then used the receiver operating characteristic (ROC) curve and the area enclosed by the abscissa as the area under the curve (AUC) value to evaluate the discrimination capacity of the model. The range of AUC values was 0–1. The closer the AUC value is to 1, the more the habitat suitability deviates from a random value and the greater the correlation between the environmental variables and the habitat classification, that is, the better the prediction effect of the model, and vice versa [43]. AUC values greater than 0.9 were indicative of very good model accuracy; 0.8–0.9 were good; 0.7–0.8 were acceptable; 0.6–0.7 were poor; and less than 0.6 were invalid [46,47].

### 2.4. Model Validation Test

To validate our model predictions, we established a ginkgo field experiment with 12 test sites, three in each of the four suitable habitats predicted by the model. To exclude the genotypic effect, we used a single clone over all the test sites through grafting. We also tried to keep soil conditions and operations as consistent as possible among all the test sites. The geographical distribution, climate conditions, and soil elements at the test sites are listed in Table 2. At each site, a randomized complete block design was used. There were five replicate blocks, each separated by at least 10 m. In late March 2018, 60 ginkgo trees were grafted at each test site, with 12 trees in each block with a row spacing of 40 cm × 60 cm. The scions came from a 30-year-old female ginkgo tree (located in Pizhou Ginkgo Seed Base, 118.05° E, 34.30° N), and the rootstocks came from 3-year-old ginkgo trees in a local plantation. The trees were irrigated during extreme droughts (less than 30% of field water holdings).

**Table 2.** Data on the geographical distribution, climate factors, and soil elements at the 12 test sites.

| Suitable Category | Site | Latitude (°N) | Longitude (°E) | Altitude (m) | MAT (°C) | MAP (mm) | N (g/kg) | P (g/kg) | C (g/kg) | K (g/kg) |
|---|---|---|---|---|---|---|---|---|---|---|
| Unsuitable habitat | 1 | 22.49 | 112.50 | 143 | 22.4 | 1955 | 1.37 | 0.50 | 3.31 | 11.05 |
| | 2 | 43.41 | 81.11 | 820 | 5.2 | 331 | 1.19 | 0.46 | 2.14 | 9.92 |
| | 3 | 40.59 | 123.53 | 30 | 7.6 | 872 | 0.50 | 0.49 | 1.00 | 14.08 |
| Low-suitable habitat | 4 | 25.52 | 103.58 | 2160 | 14.1 | 1067 | 1.17 | 0.37 | 1.39 | 22.23 |
| | 5 | 25.17 | 110.36 | 247 | 19.5 | 1725 | 1.01 | 0.43 | 2.82 | 24.82 |
| | 6 | 25.89 | 114.52 | 131 | 19.2 | 1486 | 1.25 | 0.45 | 1.25 | 21.08 |
| Medium-suitable habitat | 7 | 31.97 | 107.43 | 442 | 14.1 | 1205 | 0.87 | 0.50 | 1.25 | 16.85 |
| | 8 | 39.34 | 117.91 | 35 | 11.5 | 627 | 1.18 | 0.43 | 1.70 | 14.12 |
| | 9 | 38.32 | 113.96 | 61 | 13.0 | 532 | 0.50 | 0.41 | 1.21 | 12.66 |
| High-suitable habitat | 10 | 34.21 | 117.58 | 44 | 14.5 | 845 | 0.75 | 0.48 | 1.26 | 18.60 |
| | 11 | 32.12 | 120.51 | 10 | 15.4 | 1046 | 0.33 | 0.29 | 1.07 | 19.26 |
| | 12 | 32.90 | 113.38 | 93 | 15.4 | 905 | 1.28 | 0.61 | 2.02 | 23.51 |

Leaf-trait measurements were performed at the peak of the vegetation season during 18–25 August 2018. All fully grown leaves were scanned (YMJ-B Area Meter, Top Cloud-ARGI Inc., Hangzhou, China) from one tree randomly selected in each block, which aimed for the calculation of total leaf area (TLA). After scanning, the leaves of each tree were collected and packed separately, placed in an icebox, and brought back to the laboratory. The leaves were then placed in an oven at 70 °C for 48 h and weighed (leaf dry mass per tree, LDM). The leaf dry mass per unit area (LMA) was defined as the ratio between the LDM and TLA [36].

Analyses of the leaf-trait data were performed using one-way analysis of variance (ANOVA) in R 3.5.1. Significant differences were calculated using the least significant difference (LSD) test and defined as $p < 0.05$.

## 3. Results

### 3.1. Model Performance and Climate Variables' Contribution

Based on the results of the correlation analysis, eight climate variables (MAT, TD, MAP, DD < 0, PAS, EXT, CMD, and AHM) were selected to construct the bioclimatic model (Figure 2). The mean test AUC value of 10 repeated operations was 0.913 with a standard deviation of 0.014, suggesting a high level of accuracy of the model. The climate variables with the largest contribution rate to the model was DD < 0 (63.5%), followed by PAS (12.0%), MAP (10.3%), and TD (8.2%). Of these four variables, two were temperature variables and two were precipitation variables and accumulatively interpreted 94% of the model (Table 3). The remaining four variables contributed relatively little to the model. The Jackknife test results showed that the prediction performances of climate variables were acceptable (AUC > 0.7) except for CMD. The reliability of models constructed with different climatic variables was ranked from high to low, followed by DD > 0, MAT, PAS, MAP, EXT, TD, AHM, and CMD. The models constructed with the top four climate variables reached a good level (AUC > 0.8). Response curves (Figure 3) clearly showed that the four climate variables at a good level (DD > 0, MAT, PAS, and MAP) were in the range of 0–25 °C, 12–17 °C, 0–10 mm, and 700–2900 mm, respectively, which were more suitable for the survival of ginkgo.

**Table 3.** Contribution rate and predictive ability of the climate variables to the model.

| Variable | Percent Contribution (%) | Permutation Importance (%) | Jackknife Test AUC | Appropriate Interval (Logistic Output > 0.5) |
|---|---|---|---|---|
| DD < 0 | 63.5 | 0.8 | 0.886 | 0–25 °C |
| PAS | 12.0 | 4.9 | 0.806 | 0–10 mm |
| MAP | 10.3 | 50.8 | 0.804 | 700–2900 mm |
| TD | 8.2 | 7.0 | 0.772 | 21–28.5 °C |
| CMD | 2.2 | 3.8 | 0.698 | 225–510 |
| MAT | 2.1 | 21.1 | 0.876 | 12–17 °C |
| EXT | 1.7 | 11.7 | 0.782 | 35–38 °C |
| AHM | 0 | 0 | 0.703 | 7.5–38.5 |

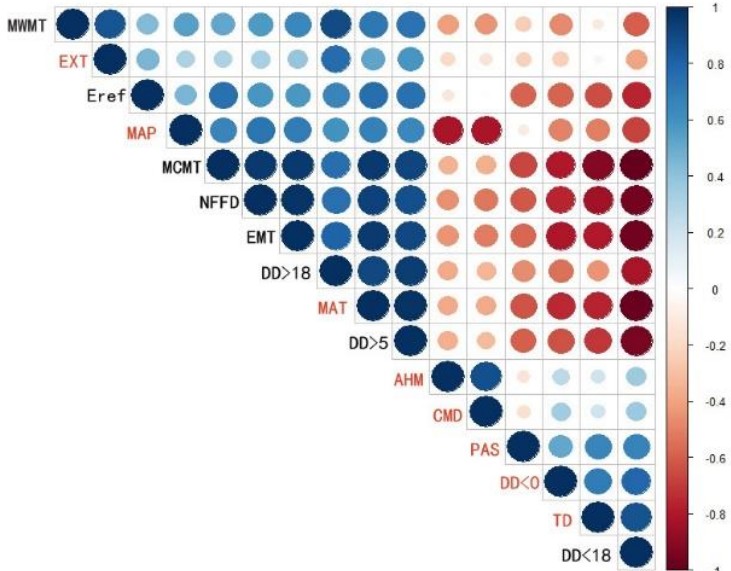

**Figure 2.** Pearson correlations of the 16 climate variables listed in Table 1. The blue circles are for positive correlations and the red ones for negative correlations. The stronger the correlation, the larger the size of the circle and the darker the color.

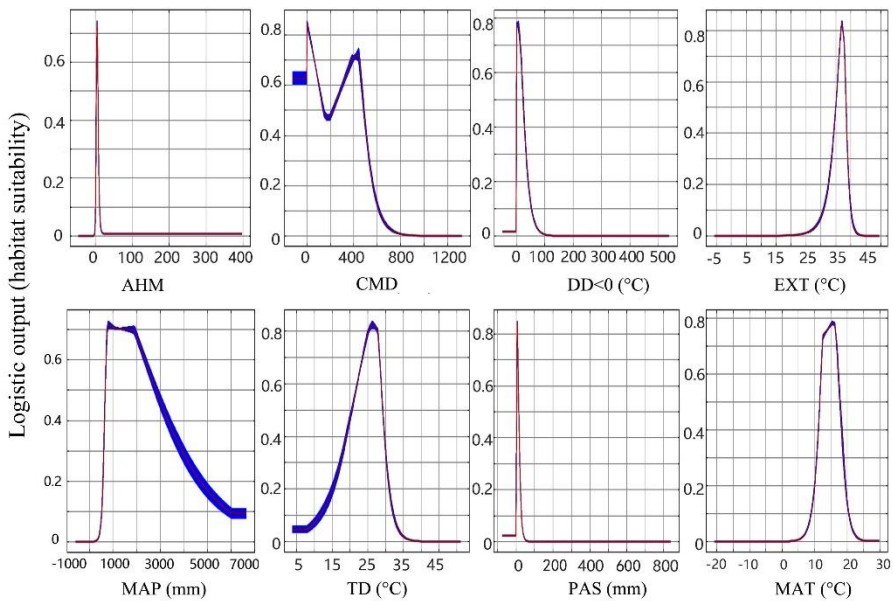

**Figure 3.** Response curves of the eight climate variables in the ginkgo bioclimatic model. The maximum entropy (MaxEnt) logistic output (also known as habitat suitability) is represented by the vertical *Y*-axis and the climate variable by the horizontal *X*-axis. When the logical output value is greater than 0.5, the probability of species presence under this climate condition is higher than that under a 'typical' condition, which indicates that the climate condition is suitable for tree species. The red curves shown are the averages over 10 replicate runs; blue margins show ±1 standard deviation (SD) calculated over 10 replicates.

### 3.2. Distribution of Current Habitat Suitability

The four categories of climatic suitability of ginkgo habitats for the current climate were mapped over entire China (Figure 4). The high-suitable habitats were concentrated in the central and eastern regions of China, accounting for 9.29% of the country's land area. The medium-suitable habitats were spread throughout the high-suitable habitats, accounting for 6.09%, and the low-suitable habitats accounted for 8.46%. The other areas were unsuitable habitats for this species, accounting for 76.16%.

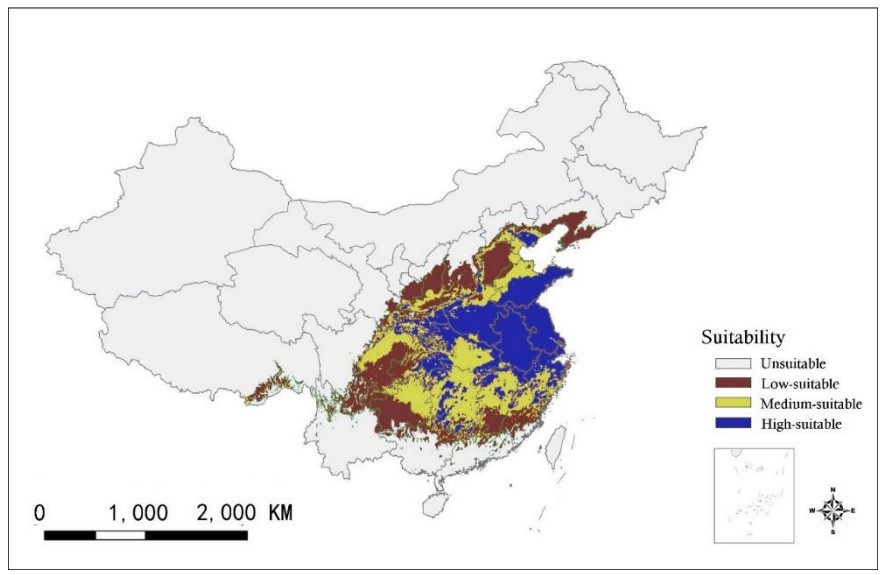

**Figure 4.** Distributions of the current (1960–1990) habitat suitability of ginkgo in China.

### 3.3. Validation of the Bioclimatic Model

The ANOVA results (Figure 5) showed that both the leaf dry mass per unit area (LMA) and leaf dry mass per tree (LDM) differed significantly among the habitat categories ($p < 0.05$). More importantly, the rankings of these two traits were consistent with the bioclimatic model predictions, suggesting the biological and economic significances of our bioclimatic model predictions. The LDMs exhibited significant differences among the high-, medium-, and low-suitable habitats, although no significant difference was identified between the low-suitable and unsuitable habitats. The LDM in high-suitable habitats was more than an order of magnitude higher than that of unsuitable habitats.

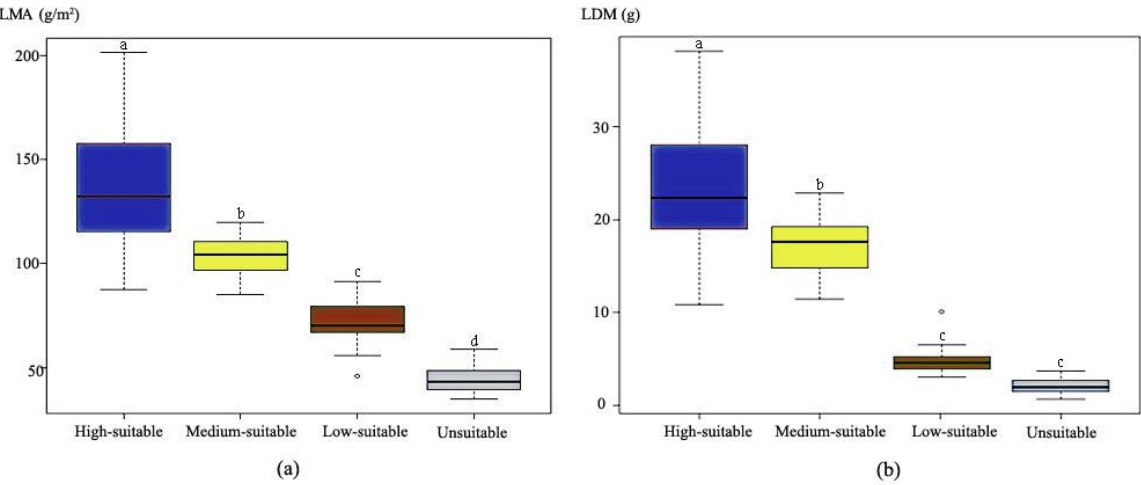

**Figure 5.** Differences in leaf-traits of ginkgo observed in the different suitable habitat categories (single-factor ANOVA). Box plots (**a**,**b**) show difference in leaf dry mass per unit area (LMA) (g/m$^2$) and leaf dry mass per tree (LDM) (g) in four types of suitable habitat, respectively. Different letters indicate significant differences among mean values ($p < 0.05$).

### 3.4. Distribution of Future Habitat Suitability

The bioclimatic model projected that the geographical distribution and area of suitable habitat for ginkgo in China would change in future periods. The suitable habitat would shift northward; however, its longitudinal shift would not be substantial. Compared with the current distribution (Figure 4), under the RCP 8.5 scenario at the end of this century, the latitudes of the northern edge of the high-, medium-, and low-suitable habitat regions would increase by 5.63°, 5.09°, and 1.68°, respectively (Figure 6). A similar but more pronounced shift would occur at the southern edge of the three types of suitable habitats.

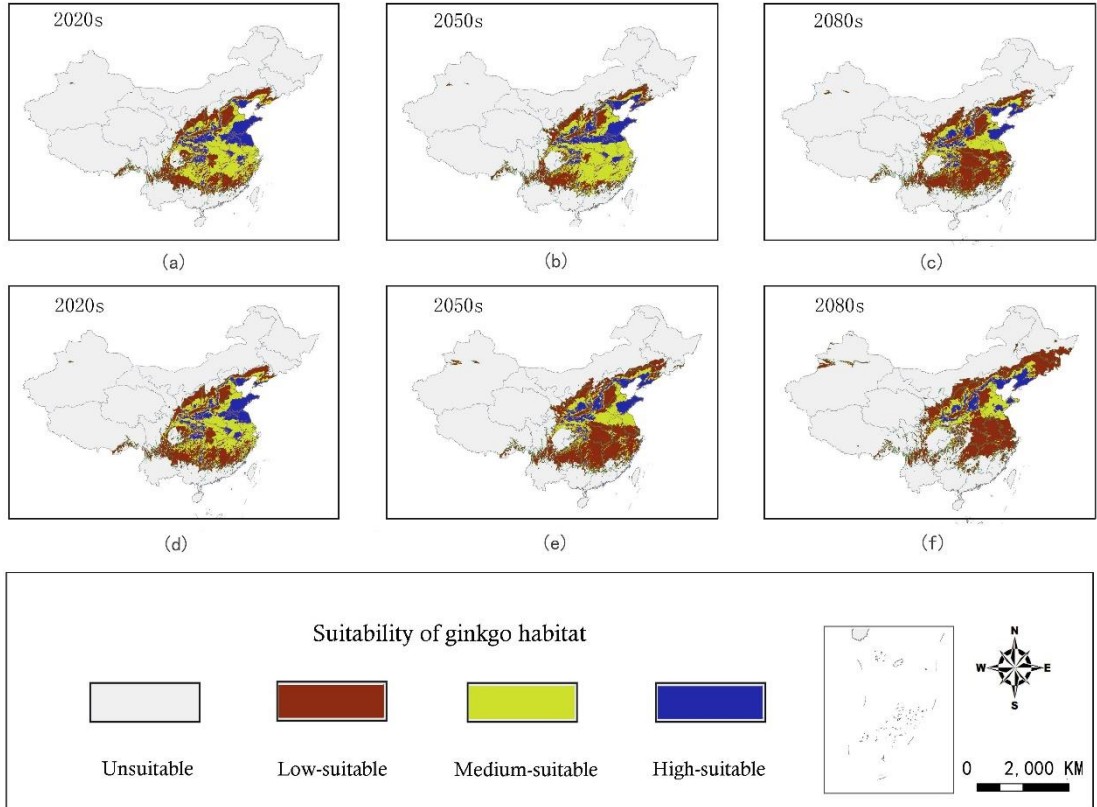

**Figure 6.** Maps of the habitat suitability of ginkgo in China in the 2020s, 2050s, and 2080s under the RCP4.5 (**a**–**c**) and the RCP 8.5 (**d**–**f**) emissions scenarios, respectively.

By the 2020s, the area of suitable habitat was predicted to change slightly under the RCP4.5 and RCP8.5 scenarios. By the 2050s, the changes in the area of suitable habitat (high- and medium-suitable) under the RCP4.5 and RCP8.5 scenarios were predicted to differ under the RCP 4.5 scenario; the area of suitable habitat would increase by 3.24% (Figure 7b). However, under the RCP 8.5 scenario, the area would be reduced by 36.77%, as most medium-suitable habitats would be transformed into the low-suitable category (Figure 7e). By the 2080s, the area of suitable habitat would be decreased significantly (37.43% and 49.58%, respectively) in the two climatic scenarios (Figure 7c,f). It is noteworthy that the area of unsuitable habitat would also decrease at the end of this century, with decreases of 1.76% and 2.09%, respectively, under the RCP 4.5 and RCP 8.5 scenarios.

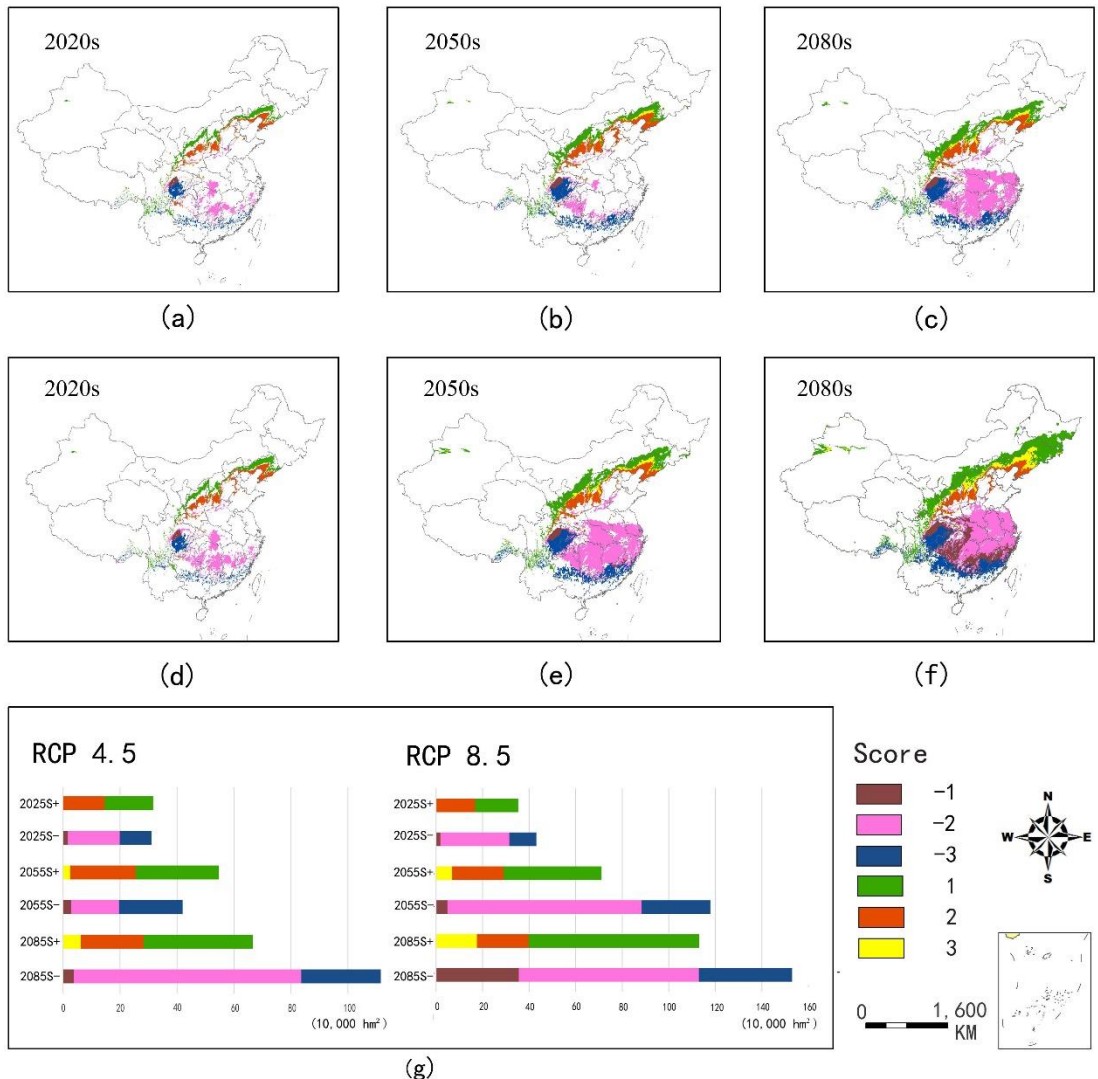

**Figure 7.** Area changes in different suitable habitats types of ginkgo relative to the current by the 2020s, 2050s, and 2080s under RCP4.5 (**a**–**c**) and RCP8.5 (**d**–**f**), respectively. (**g**) Summary table. −1 = low-suitable habitat becomes unsuitable habitat; −2 = medium -suitable and high-suitable habitat become low-suitable habitat; −3 = medium-suitable and high-suitable habitat become unsuitable habitat; 1 = unsuitable habitat become low-suitable habitat; 2 = low-suitable habitat become mid- suitable and high-suitable habitat; and 3 = unsuitable habitat become medium-suitable and high-suitable habitat.

## 4. Discussion

It is important to understand and quantify the response of ginkgo, an economically and ecologically important tree species, to a rapidly changing climate. A high-accuracy bioclimatic model was built in this study (AUC = 0.913, SD = ±0.114), and the most important climate variable (DD < 0) was identified. Model predictions were used to classify habitat suitability in biogeography, which was validated by field tests with high consistency. Future projections suggested that the geographical distribution of suitable habitats for this species would shift northward under climate change, accompanied by a certain degree of decrease in the area of suitable habitats. This information would be critical for developing adaptive strategies for resource management and development planning of this species.

### 4.1. Key Climate Factors Determining the Ginkgo Distribution

The bioclimatic model in this study was based on the hypothesis that the distribution characteristics of ginkgo on a continental scale were mainly determined by climate. A large number of simulation results support this hypothesis, suggesting that the distribution of many species is actually balanced with the current climate on a macro scale [22,48,49]. The climate variable DD < 0 ranked first in the model contribution rate and in individual prediction ability, and therefore, this variable was considered to be the most important factor restricting the distribution of *Ginkgo biloba*. This inference might be related to the effect of sub-zero temperatures on winter dormancy of ginkgo. Because cold temperatures regulate endodormancy releases from within plants, such temperatures are considered to be a major factor in determining a plant's performance in a given climate or habitat [50]. Notably, the 15 existing natural ginkgo populations that are of great concern to the academic community are all distributed in the south of the January zero-degree isotherm [51], further indicating that ginkgo is quite sensitive to cold temperatures [52]. Therefore, establishing large ginkgo plantations in low-temperature areas is not currently recommended. As the climate warms, introducing gingko to areas currently with a low-temperature will be possible in the future.

### 4.2. Distributions of Bioclimatic Habitat Categories

Based on our model predictions, four categories of climatic suitability of ginkgo habitats were mapped over entire China in this study (Figure 4). Under current climate conditions, both high- and medium-suitable habitats of ginkgo were centered in the central and eastern regions of China, including Shandong, Jiangsu, and Anhui Provinces, while the low-suitable habitats were distributed around the formers. Most of the other areas in China were unsuitable habitats of ginkgo (accounting for 76.16%). The intriguing finding was that both leaf dry weight per unit area (LMA) and leaf dry mass per tree (LDM) decreased along with our predicted four categories from high-suitable to unsuitable habitats in field tests. The LMA is a key trait in plant growth, which has been considered to play a central role in various plant adaptation strategy schemes [36,53]. There is little variability in the LMA among individuals of this species from the same location (4%) [54], however, climate conditions and other factors, such as light and $CO_2$ content, have an important impact on the LMA [55–58]. LDM has become an important indicator for assessing the productivity of ginkgo plantation because ginkgo leaves have been widely used as raw materials for medicines, cosmetics, and natural health care products. Our model predictions of habitat suitability classification are in accord with the understanding of ecological plant physiology and economics, which is important for guiding the selection of planting locations of ginkgo in production practice.

### 4.3. Impacts of Climate Change on the Habitat Suitability of Ginkgo in the Future

A meta-analysis of long-term trends in more than 1700 species revealed that more than half of the species showed significant changes in their phenologies and/or their distributions over the past 20 to 140 years [59]. Therefore, creating a realistic understanding of ginkgo persistence requires knowledge of adaptive potential in future climate conditions. Our model projections for future periods suggest that the suitable (high- and medium-suitable) habitats of ginkgo would shrink and shift northward under climate change. These predictions are consistent with previous studies that climate change is widely recognized to have a negative impact on trees near the warm edge of a species' distribution [60–62]. In the future, low-latitude regions in southern China, such as Guangdong, Guangxi, and Fujian Provinces will become unsuitable habitats for ginkgo. Shandong, Jiangsu and Anhui Provinces, which are high-suitable habitats of ginkgo at present, may be transformed into low-suitable habitats by the end of this century. However, as is the case for many tree species, the suitability of habitats in northern China will benefit from climate change [63,64]. Although the Xinjiang Autonomous Region is currently an unsuitable habitat for ginkgo, low-suitable habitats for ginkgo may occur in the north-western portion of Xinjiang in the future. Projections suggest that the climatic conditions in Jilin

Province and Heilongjiang Province in the north-eastern region may also gradually suitable for the growth of ginkgo in the future.

Under the RCP4.5 scenario, the increase in suitable habitat area would be more than the decrease by the 2050s. Our results suggest that moderate warming may have a positive impact on the ginkgo planting industry over a period of time. However, under extreme scenarios (RCP8.5), the suitable habitat area of ginkgo would continue to decrease, possibly by half by the end of the century, which would be a serious concern to policymakers and forest resource managers.

## 5. Conclusions

In this study, a bioclimatic model for ginkgo was built to understand the potential impacts of climate change on the habitat suitability of ginkgo. We identified major climate variables that affect the distribution of ginkgo and predicted four types of suitable habitat for ginkgo under current and projected climatic conditions, that were mapped and validated by a field test. Our future projections indicate that climate change may negatively impact the main productive areas of ginkgo in the future, although some new areas would become suitable for ginkgo production. The projections can be used to provide recommendations for cultivators and plantation managers to develop short-term and long-term adaptive strategies for forestry development planning under a rapidly changing climate.

**Author Contributions:** G.W. and T.W. conceived and designed the experiments; Y.G., J.G., and X.S. performed the experiments, analyzed the data, and wrote the manuscript; T.W. revised the manuscript.

**Funding:** This research was funded by the National Key Research and Development Program of China (2017YFD0600700), the Agricultural Science and Technology Independent Innovation Funds of Jiangsu Province (CX (16) 1005), the Doctorate Fellowship Foundation of Nanjing Forestry University.

**Conflicts of Interest:** The authors declare no conflict of interest.

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
