# Peer review of "Predicting the Bioclimatic Habitat Suitability of Ginkgo biloba L. in China with Field-Test Validations"

_forests, doi:10.3390/f10080705_

Round 1

Reviewer 1 Report

This is an interesting and useful study with relevant potential applications in forestry. The text is well-written enough to be easy to follow and pleasant to read, despite some minor grammar mistakes -- a quick check by a native English speaker could correct these, although I don’t consider this essential. However, I have a number of other (mostly also minor) issues that do need to be addressed, mainly regarding the clarification of concepts and a proper use of citations:

Ln 13: AUC and SD must be defined at first use, or not be used in the abstract. AUC can be replaced here with “discrimination capacity” or similar.

Ln 15: “heat sum below zero” is also unclear and should be defined here.

Ln 23: “move” sounds strange when applied to “habitats”; I would use “shift” instead (here and elsewhere in the manuscript).

Ln 55-57: bioclimatic models don’t always model “probability of occurrence”; more often (and also in the case of the Maxent modelling algorithm, which was used in this manuscript) they model climatic / habitat suitability, which is a mathematically different thing. Also, [20] is not an adequate reference for this statement: a more basal article or textbook on the concepts behind bioclimatic modelling should be cited.

Ln 64-65: again, this reference is not appropriate because it does not show that generally Maxent “can accurately predict species distributions”. A more thorough comparative study/studies should be cited here. See also comment on Ln 71-72 below.
Ln 66-67: it is misleading to say that Maxent uses “presence-only data” without explicitly adding that it also requires a modelling background (within a sensible study area) against which to compare the environmental values of the presence points. Please change this to “presence-background data”, or at least mention the background in the next line, e.g. “and categorical and continuous data (FOR THE PRESENCE POINTS AND A RELEVANT BACKGROUND)”.

Ln 68-69: you cannot say that “its prediction accuracy is ALWAYS stable and reliable” (I have seen several instances in which it was not), and again reference 27 does not show such generality. See also next comment.

Ln 71-72: I agree with this statement, but it contradicts two sentences in the previous paragraph about Maxent prediction accuracy (ln 65, ln 68). Maybe rephrase the previous statements to say that Maxent generally provides good discrimination ability within the modelled region -- which is not really “predicting” something else, such as distributional changes or physiological performance.

Ln 107: thinning occurrence points does not necessarily “avoid spatial autocorrelation” -- to test this you would need to compute a spatial correlogram and check from which distance the points stop being autocorrelated for a given variable(s). Not that I think this is necessary, as spatial autocorrelation is a natural and necessary process in natural systems, and it can’t really (and probably shouldn’t) be “avoided”. Thinning can, however, be useful to reduce the clustering of surveyed occurrence points.

Ln 130-132: Pearson correlation coefficients measure pairwise (rather than multiple) correlations among variables, so they are not a “multicollinearity test” (which could be done e.g. using the variance inflation factor, although that’s not necessarily required). Also, you should not eliminate “variables with a correlation coefficient greater than 0.8”, but rather one variable (preferably the least important one) within each pair of variables correlated above the threshold.

Ln 153 and elsewhere: the AUC does not really measure “accuracy” (which would imply the precision of the continuous predicted values), but rather “discrimination capacity” (i.e., the ability of these values to distinguish presence from background points).

Ln 153-154: AUC values of 0.5 actually correspond to random predictions, so it’s not “the larger the AUC value” but rather “the farther away from 0.5”, or (probably better) “the closer to 1”. Values close to 0 indicate that the model is discriminating well (i.e. separating presences from background, rather than mixing them randomly), but classifying the points in the opposite way.

Ln 205 (Fig 3): The Maxent logistic output tries to estimate / approximate, but is not mathematically, a “probability of presence”. Please just call it “habitat suitability”, “Maxent logistic output” or “Maxent logistic prediction”, but not probability. This applies to the figure axis, figure caption, and anywhere else in the manuscript.

Ln 219 (Fig 4): It’s probably just a matter of habit, but I find it counter-intuitive that red means high and green means low suitability for the species. More importantly, these colours are undistinguishable for colour-blind readers (who are a lot). Please consider using a colour-blind-friendly colour scheme, e.g. with blue instead of green -- there are numerous examples online.

Fig 5: “LAM” should be “LMA”

Ln 235: Please replace “The distribution” (which can be misleading) with “Suitable habitats” or similar.

Fig 6: I can’t see the years in the figure or legend.

Ln 244: add “of suitable habitat” (or similar) after “the area”

Reviewer 2 Report

This is a great effort and a great manuscript with a reasonable flow of ideas and an excellent experimental design and procedures.

The work and objectives are quite timely and were well conceived.

There are however some concerns ranging from simple that can easily be addressed to some more complex that need some elaborate and convincing clarifications.

The simple notes are:

Why the 4x4 km2 areas, where does the 4x4 km2 come from? any justifications? The 800x800m pixel size was also a bit confusing. I realize it is a number at the end but why exactly 800x800 m, why not the more standard units of 1kmx1km or 500mx500m, etc... 800 sound sa bit odd? and I could read/find any justifications? The bioclimatic model was not very well described/clarified in the manuscript  The manuscript speaks to "full expert users" which is not the general readership you should aim for.  I request that more clarifications on the model itself, parameters, performance, sensitivity, et... references be added to make it easier on readers  understand the extent of what you did. The use of abbreviation was not very successful or consistent.  Always apply the rule of "first use", spell out the name then establish the abbreviation and use form there on.  The manuscript is full of confusing and undefined abbreviations You used LAM abbreviation then LMA? I also suggest you change the color of habitat suitability to High suitability= GREEN, and low Suitability = RED, and medium YELLOW.

Generally, the work is very clear but some passages needs some elaborate clarifications and additional explanations.

On Page #8 in the validation of the bioclimatic model section, I find the work, field test, and results rather weak and very tenuous for many reasons like:

limited tests limited time span (was not too clear) limited indicators (mass/area or dry mass)

I am not certain I agree 100% with the validity of these results based on the very limited work and the selection of one genetic background. Using more than one genetic background would have been more convincing as adaptability/resilience means genetic change to a degree.

I hope you can tone down the meaning of that part of the work and focus on the model results (they are great in themselves) and not directly link them to the limited field tests/results you reported.

Please consult the annotated PDF file for further comments, notes, corrections, suggestions, etc... 
